# Text-to-Image Diffusion Models are Zero-Shot Classifiers

**Kevin Clark**[*]
Google DeepMind
Toronto
kevclark@google.com

**Priyank Jaini**[*]
Google DeepMind
Toronto
pjaini@google.com

## Abstract

The excellent generative capabilities of text-to-image diffusion models suggest they learn informative representations of image-text data. However, what knowledge their representations capture is not fully understood, and they have not been thoroughly explored on downstream tasks. We investigate diffusion models by proposing a method for evaluating them as zero-shot classifiers. The key idea is using a diffusion model's ability to denoise a noised image given a text description of a label as a proxy for that label's likelihood. We apply our method to Stable Diffusion and Imagen, using it to probe fine-grained aspects of the models' knowledge and comparing them with CLIP's zero-shot abilities. They perform competitively with CLIP on a wide range of zero-shot image classification datasets. Additionally, they achieve state-of-the-art results on shape/texture bias tests and can successfully perform attribute binding while CLIP cannot. Although generative pre-training is prevalent in NLP, visual foundation models often use other methods such as contrastive learning. Based on our findings, we argue that generative pre-training should be explored as a compelling alternative for vision-language tasks.

## 1 Introduction

Large models pre-trained on internet-scale data can adapt effectively to a variety of downstream tasks. Increasingly, they are being used as zero-shot learners with no task-specific training, such as with CLIP (Radford et al., 2021) for images and GPT-3 (Brown et al., 2020) for text. In natural language processing, many successful pre-trained models are generative (i.e., language models). However, generative pre-training is less commonly used for visual tasks. Until recently, the usual practice for vision problems was to pre-train models on labeled datasets such as Imagenet (Deng et al., 2009), or JFT (Sun et al., 2017). Later research in visual and vision-language problems has led to image-text models pre-trained primarily using either contrastive losses (Radford et al., 2021; Jia et al., 2021; Yuan et al., 2021) or autoencoding tasks (Vincent et al., 2010; He et al., 2022).

On the other hand, generative text-to-image models based on denoising diffusion probabilistic models (Ho et al., 2020) such as Imagen (Saharia et al., 2022a), Dalle-2 (Ramesh et al., 2022), and Stable Diffusion (Rombach et al., 2022) can generate realistic high-resolution images and generalize to diverse text prompts. Their strong performance suggests that they learn effective representations of image-text data. However, their ability to transfer to downstream discriminative tasks and how they compare to other pre-trained models has not been explored thoroughly.

In this paper, we investigate these questions by transferring Imagen and Stable Diffusion (SD) to discriminative tasks. While previous studies have used representations from diffusion models for downstream tasks (Brempong et al., 2022; Burgert et al., 2022; Zhao et al., 2023), we instead propose

---

[*]equal contribution

37th Conference on Neural Information Processing Systems (NeurIPS 2023).

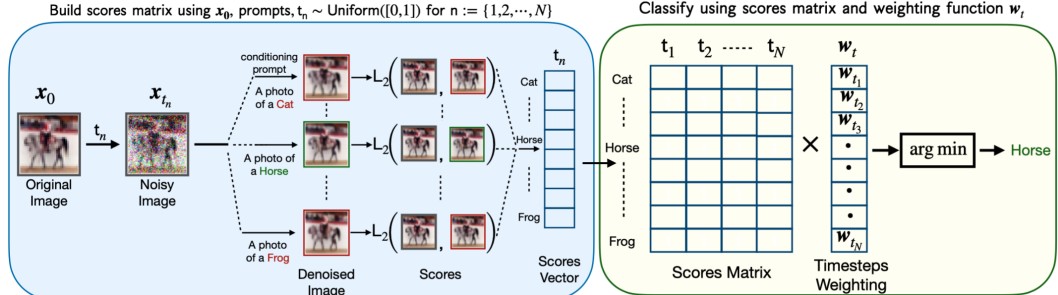

Figure 1: **Zero-Shot Classification using Diffusion Models.** We first compute denoising scores for each label prompt across multiple time-steps to generate a scores matrix. We then classify an image by aggregating the scores for each class using a weighting function over the time-steps. The image is assigned the class with the minimum aggregate score. In Section 3.1, we discuss how efficiency can be improved only computing a subset of the full scores matrix.

a way of using text-to-image diffusion models directly as zero-shot image classifiers. Our method essentially runs the models as generative classifiers (Ng & Jordan, 2001), using a re-weighted version of the variational lower bound to score images since diffusion models do not produce exact likelihoods. More specifically, the method repeatedly noises and denoises the input image while conditioning the model on a different text prompt for each possible class. The class whose text prompt results in the best denoising ability is predicted. This procedure is expensive because it requires denoising many times per class (with different noise levels). To make it usable in practice, we present improvements that increase the method's sample efficiency by up to 1000x, such as pruning obviously-incorrect classes early. While still requiring too much compute to be an easily-deployable classifier, our method allows us to quantitatively study fine-grained aspects of a diffusion model's learned knowledge through evaluation on classification tasks (as opposed to qualitatively examining model generations).

We compare Imagen and SD against CLIP[2] (Radford et al., 2021), a widely used model for zero-shot image-text tasks trained with contrastive learning. A high-level goal of the experiments is to see the strengths and weaknesses of generative and contrastive pre-training for computer vision. First, we demonstrate that diffusion models have strong zero-shot classification accuracies (competitive with CLIP) on several diverse vision datasets. Next, we show both Imagen and SD performs remarkably well on the Cue-Conflict dataset (Geirhos et al., 2019), where images have been stylized with textures conflicting with their labels. For example, Imagen achieves >50% error reduction over CLIP and even outperforms the much larger ViT-22B (Dehghani et al., 2023) model. This finding is particularly interesting because, unlike supervised classifiers, humans are known to be much more reliant on shape than texture when identifying images. Lastly, we study attribute binding using the synthetic data from Lewis et al. (2022), and find that, unlike CLIP, diffusion models can successfully bind together attributes in some settings.

The main contributions of this paper are:

- We show text-to-image diffusion models can be used as effective zero-shot classifiers. While using too much compute to be very practical on downstream tasks, the method provides a way of quantitatively studying what the models learn.

- We develop techniques that hugely lower the compute cost of these zero-shot classifiers, making them usable (although still slow) on datasets with many classes.

- We demonstrate the strong generalization capabilities of Imagen and Stable Diffusion, resulting in good zero-shot performance on vision datasets (comparable to CLIP).

- We show that diffusion models are robust to misleading textural cues, achieving state-of-the-art results on Cue-Conflict.

- We use our framework to study attribute binding in diffusion models and find that they can perform some binding tasks while CLIP cannot.

---

[2]We use ViT-L/14, the largest public CLIP model

Together, our results suggest that text-to-image diffusion models learn powerful representations that can effectively be transferred to tasks beyond image generation.

## 2 Preliminaries

Diffusion models (Sohl-Dickstein et al., 2015; Ho et al., 2020; Song et al., 2020; Song & Ermon, 2020) are latent variable generative models defined by a forward and reverse Markov chain. Given an unknown data distribution, $q(\boldsymbol{x}_0)$, over observations, $\boldsymbol{x}_0 \in \mathbb{R}^d$, the forward process corrupts the data into a sequence of noisy latent variables, $\boldsymbol{x}_{1:T} := \{\boldsymbol{x}_1, \boldsymbol{x}_2, \cdots, \boldsymbol{x}_T\}$, by gradually adding Gaussian noise with a fixed schedule defined as:

$$q(\boldsymbol{x}_{1:T}|\boldsymbol{x}_0) := \prod_{t=1}^{T} q(\boldsymbol{x}_t|\boldsymbol{x}_{t-1}) \tag{1}$$

where $q(\boldsymbol{x}_t|\boldsymbol{x}_{t-1}) := \mathsf{Normal}(\boldsymbol{x}_t; \sqrt{1 - \beta_t}\boldsymbol{x}_{t-1}, \beta_t \boldsymbol{I})$. The reverse Markov process gradually denoises the latent variables to the data distribution with learned Gaussian transitions starting from $\mathsf{Normal}(\boldsymbol{x}_T; 0, \boldsymbol{I})$ i.e.

$$p_{\boldsymbol{\theta}}(\boldsymbol{x}_{0:T}) := p(\boldsymbol{x}_T) \cdot \prod_{t=0}^{T-1} p_{\boldsymbol{\theta}}(\boldsymbol{x}_{t-1}|\boldsymbol{x}_t)$$

$p_{\boldsymbol{\theta}}(\boldsymbol{x}_{t-1}|\boldsymbol{x}_t) := \mathsf{Normal}\big(\boldsymbol{x}_{t-1}; \boldsymbol{\mu}_{\boldsymbol{\theta}}(\boldsymbol{x}_t, t), \boldsymbol{\Sigma}_{\boldsymbol{\theta}}(\boldsymbol{x}_t, t)\big)$. The aim of training is for the forward process distribution $\{\boldsymbol{x}_t\}_{t=0}^T$ to match that of the reverse process $\{\tilde{\boldsymbol{x}}_t\}_{t=0}^T$ i.e., the generative model $p_{\boldsymbol{\theta}}(\boldsymbol{x}_0)$ closely matches the data distribution $q(\boldsymbol{x}_0)$. Specifically, these models can be trained by optimizing the variational lower bound of the marginal likelihood (Ho et al., 2020; Kingma et al., 2021):

$$-\log p_\theta(\boldsymbol{x}_0) \leq -\mathsf{VLB}(\boldsymbol{x}_0) := \mathcal{L}_{\mathsf{Prior}} + \mathcal{L}_{\mathsf{Recon}} + \mathcal{L}_{\mathsf{Diffusion}}$$

$\mathcal{L}_{\mathsf{Prior}}$ and $\mathcal{L}_{\mathsf{Recon}}$ are the prior and reconstruction loss that can be estimated using standard techniques in the literature (Kingma & Welling, 2014). The (re-weighted) diffusion loss can be written as:

$$\mathcal{L}_{\mathsf{Diffusion}} = \mathbb{E}_{\boldsymbol{x}_0, \boldsymbol{\varepsilon}, t}\Big[\boldsymbol{w}_t\|\boldsymbol{x}_0 - \tilde{\boldsymbol{x}}_{\boldsymbol{\theta}}(\boldsymbol{x}_t, t)\|_2^2\Big]$$

with $\boldsymbol{x}_0 \sim q(\boldsymbol{x}_0)$, $\boldsymbol{\varepsilon} \sim \mathsf{Normal}(0, \boldsymbol{I})$, and $t \sim \mathcal{U}([0, T])$. Here, $\boldsymbol{w}_t$ is a weight assigned to the timestep, and $\tilde{\boldsymbol{x}}_{\boldsymbol{\theta}}(\boldsymbol{x}_t, t)$ is the model's prediction of the observation $\boldsymbol{x}_0$ from the noised observation $\boldsymbol{x}_t$. Diffusion models can be conditioned on additional inputs like class labels, text prompts, segmentation masks or low-resolution images, in which case $\tilde{\boldsymbol{x}}_{\boldsymbol{\theta}}$ also takes a conditioning signal $\boldsymbol{y}$ as input.

## 3 Zero-Shot Classification using Diffusion Models

In this section, we show how to convert the generation process of a text-to-image diffusion model into a zero-shot classifier to facilitate quantitative evaluation on downstream tasks. Figure 1 shows an overview of our method.

**Diffusion Generative Classifier:** We begin with a dataset, $\big\{(\boldsymbol{x}^1, y^1), \ldots, (\boldsymbol{x}^n, y^n)\big\} \subseteq \mathbb{R}^{d_1 \times d_2} \times [\mathsf{y}_K]$ of $n$ images[3] where each image belongs to one of $K$ classes $[\mathsf{y}_K] := \{\mathsf{y}_1, \mathsf{y}_2, \cdots, \mathsf{y}_K\}$. Given an image $\boldsymbol{x}$, our goal is to predict the most probable class assignment

$$\tilde{y} = \underset{\mathsf{y}_k}{\arg\max}\, p(y = \mathsf{y}_k|\boldsymbol{x}) = \underset{\mathsf{y}_k}{\arg\max}\, p(\boldsymbol{x}|y = \mathsf{y}_k) \cdot p(y = \mathsf{y}_k) = \underset{\mathsf{y}_k}{\arg\max}\, \log p(\boldsymbol{x}|y = \mathsf{y}_k).$$

where we assume a uniform prior $p(y_i = \mathsf{y}_k) = \frac{1}{k}$ that can be dropped from the $\arg\max$.[4] A generative classifier (Ng & Jordan, 2001) uses a conditional generative model with parameters $\theta$ to estimate the likelihood as $p_\theta(\boldsymbol{x}|y = \mathsf{y}_k)$.

Using a text-to-image diffusion model as a generative classifier requires two modifications. First, the models are conditioned on text prompts rather than class labels. Thus we convert each label, $\mathsf{y}_k$, to text using a mapping $\phi$ with a dataset-specific template (e.g. $\mathsf{y}_k \to \mathsf{A\ photo\ of\ a\ }\mathsf{y}_k$). Second,

---

[3]For simplicity, we use $\boldsymbol{x}$ in place of $\boldsymbol{x}_0$ to refer to an image.

[4]We can't use a learned prior in the zero-shot setting.

diffusion models do not produce exact log-likelihoods (i.e. we cannot compute $\log p_\theta(\boldsymbol{x}|y = y_k)$ directly). Our key idea for a solution is to use the VLB (more specifically $\mathcal{L}_{\text{Diffusion}}$ as Imagen and SD are not trained with the other losses) as a proxy. Thus we have:

$$\tilde{y} = \underset{y_k}{\arg\max} \ \log p_\theta(\boldsymbol{x}|y = y_k) \approx \underset{y_k}{\arg\min} \ \mathcal{L}_{\text{Diffusion}}(\boldsymbol{x}, y_k)$$

$$= \underset{y_k \in [y_K]}{\arg\min} \ \mathbb{E}_{\epsilon,t}\left[\boldsymbol{w}_t \|\boldsymbol{x} - \tilde{\boldsymbol{x}}_{\boldsymbol{\theta}}(\boldsymbol{x}_t, \boldsymbol{\phi}(y_k), t)\|_2^2\right] \quad (2)$$

Note that for SD, $\boldsymbol{x}$ and $\tilde{\boldsymbol{x}}_{\boldsymbol{\theta}}$ are latent representations, with $\boldsymbol{x}$ obtained by encoding the image using a VAE. With Imagen on the other hand, $\boldsymbol{x}$ consists of the raw image pixels.

**Estimating the Expectation:**  We approximate the expectation in Equation (2) using Monte-Carlo estimation. At each step, we sample a $t \sim \mathcal{U}([0, 1])$ and then a $\boldsymbol{x}_t$ according to the forward diffusion process (Equation (1)): $\boldsymbol{x}_t \sim q(\boldsymbol{x}_t|\boldsymbol{x}_0)$. Next, we denoise this noisy image using the model (i.e. we use it to predict $\boldsymbol{x}$ from $\boldsymbol{x}_t$), obtaining $\hat{\boldsymbol{x}} = \tilde{\boldsymbol{x}}_{\boldsymbol{\theta}}(\boldsymbol{x}_t, \boldsymbol{\phi}(y_k), t)$. We call the squared error of the prediction, $\|\boldsymbol{x} - \hat{\boldsymbol{x}}\|_2^2$, a *score* for $(\boldsymbol{x}, y_k)$. We score each class $N$ times, obtaining a $K \times N$ *scores matrix*[5] for the image. Finally, we weight the scores according to the corresponding $\boldsymbol{w}_t$ and take the mean, resulting in an estimate of $\mathcal{L}_{\text{Diffusion}}$ for each class.

**Choice of Weighting Function:**  Imagen and SD are trained with the "simple" loss, where $\boldsymbol{w}_t = \text{SNR}(t)$, the signal-to-noise ratio (Kingma et al., 2021) for timestep $t$. However, we found other weighting functions can improve results. First, we experimented with learning $\boldsymbol{w}_t$ by binning the times into 20 buckets and training a 20-features logistic regression model to learn weights for the buckets that maximize classification accuracy. However, using that weighting is not truly zero-shot since it requires label information to learn. We thus, also handcrafted a weighting function that can be used across datasets. We designed $\boldsymbol{w}_t$ by finding a simple function that looked close to our learned weighting function on CIFAR-100 (we did not look at other datasets to preserve zero-shot protocol). Interestingly, we found that the simple function $\boldsymbol{w}_t := \exp(-7t)$ works well for both Imagen an SD across tasks and used it for our experiments. As it is monotonic, $\mathcal{L}_{\text{Diffusion}}$ with this weighting can still be viewed as a likelihood-based objective that maximizes-the variational lower bound under simple data augmentations (Kingma & Gao, 2023). We provide details on learning $\boldsymbol{w}_t$ and an empirical comparison of different weighting functions in Appendix B.

### 3.1   Improving Efficiency

Computing $\tilde{y}$ with naive Monte-Carlo estimation can be expensive because $\mathcal{L}_{\text{Diffusion}}$ has fairly high variance. Here, we propose techniques that reduce the compute cost of estimating the $\arg\min$ over classes. The key idea is to leverage the fact that we only need to compute the $\arg\min$ and do not require good estimates of the actual expectations.

**Shared Noise:**  Differences between individual Monte-Carlo samples from $\mathcal{L}_{\text{Diffusion}}$ can of course be due to different $t$ or forward diffusion samples from $q(\boldsymbol{x}_t|\boldsymbol{x}_{t-1})$, whereas we are only interested in the effect of the text conditioning $\boldsymbol{\phi}(y_k)$. We find far fewer samples are necessary when we use the *same* $t$ and $\boldsymbol{x}_t$ across different classes, as shown in Figure 1. After sampling a $t \sim \mathcal{U}([0, 1])$ and $\boldsymbol{x}_t \sim q(\boldsymbol{x}_t|\boldsymbol{x}_0)$, we score all classes against this noised image instead of a single one. As a result, the differences between these estimates are only due to the different text conditioning signals.

**Candidate Class Pruning:**  Rather than using the same amount of compute to estimate the expectation for each class, we can further improve efficiency by discarding implausible classes early and dynamically allocating more compute to plausible ones. In particular, we maintain a set of candidate classes for the image being classified. After collecting a new set of scores for each candidate class, we discard classes that are unlikely to become the lowest-scoring (i.e. predicted) class with more samples. Since we are collecting paired samples (with the same $t$ and $\hat{\boldsymbol{x}}_{i,t}$), we use a paired student's t-test to identify classes that can be pruned. This pruning can be viewed as a succesive elimination algorithm for best-arm identification in a multi-armed bandit setting (Paulson, 1964; Even-Dar et al., 2002). Of course, scores do not exactly follow the standard assumptions of a student's t-test, so we

---

[5]Later we discuss how we can avoid computing the full matrix for efficiency.

use a small p-value ($2e^{-3}$ in our experiments) and ensure each class is scored a minimum number of times (20 in our experiments) to minimize the chance of pruning the correct class. The full procedure is shown in Algorithm 1.

**Comparison:** Figure 2 compares the number of samples needed to accurately classify CIFAR-100 images for different efficiency strategies. Using shared noise and pruning greatly improves efficiency, requiring up to 1000x less compute than naïve scoring. Nevertheless, classifying with a diffusion model still typically takes 10s of scores per class on average, making the diffusion classifier expensive to use for datasets with many classes.

# 4 Empirical Analysis and Results

In this section, we detail our analysis for the diffusion zero-shot classifiers on a variety of tasks. These include classification on various vision datasets to study generalization capabilities on diverse domains, evaluating model robustness to conflicting cues between texture and shape, and studying attribute binding ability through targeted evaluation on synthetic data.

We mainly compare Imagen and SD with CLIP (Radford et al., 2021). We chose CLIP because it is a well-studied and widely-used model, as our primary aim is to study diffusion models rather than push state-of-the-art zero-shot accuracies. Our experiments reveal strengths and weaknesses of image-text representations learned via generative training vs. CLIP's contrastive training.

**Model details:** Imagen is a cascaded diffusion model (Ho et al., 2022) consisting of a $64 \times 64$ low-resolution model and two super-resolution models. We only use the $64 \times 64$ model for our experiments because we found the high-resolution models performed poorly as classifiers. Combining the $64 \times 64$ model's scores with scores from the higher-resolutions models did not improve results either (see Appendix E for details). The issue is that high-resolution models condition strongly on their low-resolution inputs and are therefore less sensitive to the text prompt. Unlike with Figure 3, high-resolution denoising with different text prompts produces images imperceptibly different to the human eye because they all agree with the same low resolution image.

We use version 1.4 of Stable diffusion for our experiments. It uses a pre-trained text encoder from CLIP to encode the text and a pre-trained variational autoencoder to map images to a latent space.

CLIP consists of vision and text transformers trained with contrastive learning. We use the largest CLIP model (ViT-L/14@224px). We provide more details on all the models in Appendix A.

---

**Algorithm 1** Diffusion model classification with pruning.

---

**given**: Example to classify $\boldsymbol{x}$, diffusion model w/ params $\theta$, weighting function $\boldsymbol{w}$, hyperparameters min_scores, max_scores, cutoff_pval.

//Map from classes to diffusion model scores.
scores = $\{y_i : [] $ for $y_i \in [y_K]\}$
$n = 0$
**while** $|\text{scores}| > 1$ **and** $n < $ max_scores:
  $n = n + 1$
  //Noise the image
  $t \sim \mathcal{U}([0,1])$
  $\boldsymbol{x}_t \sim q(\boldsymbol{x}_t|\boldsymbol{x})$
  //Score against the remaining classes.
  **for** $y_i \in$ scores:
    add $\boldsymbol{w}_t \|\boldsymbol{x} - \tilde{\boldsymbol{x}}_{\boldsymbol{\theta}}(\boldsymbol{x}_t, \boldsymbol{\phi}(y_i), t)\|_2^2$
    to scores[$y_i$]
  //Prune away implausible classes.
  $\tilde{y} = \arg\min_{y_i}$ scores[$y_i$].mean()
  **if** $n \geq$ min_scores:
    **for** $y_i \in$ scores:
      **if** paired_ttest_pval(
      scores[$\tilde{y}$], scores[$y_i$]) < cutoff_pval:
        remove $y_i$ from scores.
**return** $\tilde{y}$

---

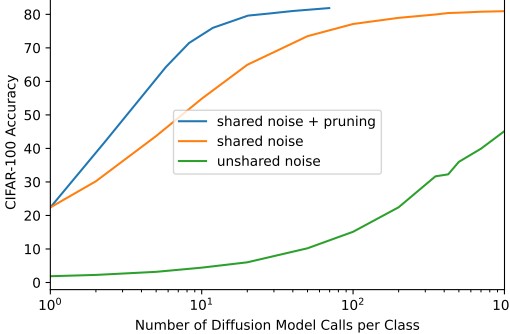

Figure 2: Comparison of efficiency improvements for Imagen on CIFAR-100. Shared noise improves sample efficiency by roughly 100x and pruning by an additional 8-10x.

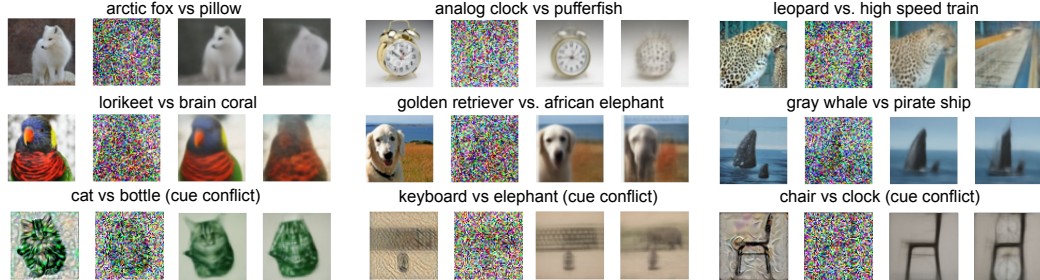

Figure 3: Example predictions from Imagen when denoising the same image with different text prompts. Each set of images shows the original, noised, and denoised images for the two classes. The top two rows use ImageNet images and the bottom row uses Cue-Conflict.

**Experiment details:** For each experiment, we obtain scores using the heuristic timestep weighting function and the efficient scoring method in Algorithm 1. Due to the method's still-substantial compute cost, we use reduced-size datasets (4096 examples) for our experiments. We preprocess each dataset by performing a central crop and then resizing the images to $64 \times 64$ resolution for Imagen, $512 \times 512$ for SD, and $224 \times 224$ for CLIP. We use min_scores $= 20$, max_scores $= 2000$, and cutoff_pval $= 2 \times e^{-3}$. These are simply reasonable choices that keep the method efficient to run; changing the values does effect the model behavior in expectation but trades off compute for reduction in variance. We use a single prompt for each image. While we could have used an ensemble of prompts (i.e, made the expectation in Equation (2) also over different prompt templates), we chose not to for the sake of simplicity, as our goal is to better understand models rather than achieve state-of-the-art zero-shot performance. Therefore, our reported results are slightly lower than in the CLIP paper, which uses prompt ensembling. We found in our experiments that diffusion models were quite robust to the choice of prompt. For example, we tried four different prompts from the CLIP templates for CIFAR-100 and found accuracies to all be within 1.5% of each other.

**Comparing models:** Imagen, SD and CLIP have different model sizes, input resolutions, and are trained on different datasets for different amounts of time, so the comparison is not direct. While ideally we would train models of the same size on the same data, this would be very expensive and challenging in practice; we instead use these strong existing pre-trained models. Our comparisons are geared towards highlighting the strengths and weaknesses of text-image diffusion models.

## 4.1 Image Classification

**Setup:** We first evaluate the performance at zero-shot classification. For this purpose, we consider 13 datasets from Radford et al. (2021) as reported in Table 1. We use the prompt templates and class labels used by Radford et al. (2021), which renames some classes that confuse models (e.g. "crane $\rightarrow$ "crane bird"" in Imagenet) (OpenAI, 2021b). We use the first prompt from the list, except for Imagenet, where we use "A bad photo of a *label* " since this is a good prompt for Imagen, SD and CLIP (OpenAI, 2021a).

Since we use the low-resolution Imagen model, we obtain results using CLIP under two settings for a more thorough comparison. In the first setting, we resize all the datasets to $64 \times 64$ which serves as the base low-resolution dataset. Imagen uses this dataset directly. For CLIP, we subsequently upsample the images, resizing them to $224 \times 224$ resolution. In the second setting, we directly resize all datasets to $224 \times 224$ resolution to obtain the best results possible using CLIP where it can take advantage of its higher input resolution.

**Results:** Results are shown in Table 1. The first eight datasets (up through EuroSAT) on the top block of the table are all originally of resolution $64 \times 64$ or less. On these datasets, Imagen generally outperforms CLIP and Stable Diffusion on classification accuracy under the same evaluation setting i.e., the models are conditioned on the same text prompts, etc. Imagen significantly outperforms CLIP on SVHN and SD on digit recognition datasets like MNIST and SVHN, which requires recognizing text in an image. Saharia et al. (2022b) observe that Imagen is particularly good at generating text,

| Model | CIFAR10 | CIFAR100 | STL10 | MNIST | DTD | Camelyon | SVHN | EuroSAT |
|---|---|---|---|---|---|---|---|---|
| Imagen | **96.6** | **84.3** | **99.6** | **79.2** | 37.3 | **60.3** | **62.7** | **60.3** |
| Stable Diffusion | 72.1 | 45.3 | 92.8 | 19.1 | **44.6** | 51.3 | 13.4 | 12.4 |
| CLIP/ViT-L/14 | 94.7 | 68.6 | 99.6 | 74.3 | 36.0 | 58.0 | 21.5 | 58.0 |

| Model | Stanford Cars | Imagenet | Caltech 101 | Oxford Pets | Food101 |
|---|---|---|---|---|---|
| Imagen | **81.0** | 62.7 | 68.9 | 66.5 | 68.4 |
| Stable Diffusion | 77.8 | 61.9 | 73.0 | 72.5 | 71.6 |
| CLIP/ViT-L/14 | 62.8/75.8 | 63.4/**75.1** | 70.2/**84.1** | 76.0/**89.9** | 83.9/**93.3** |

Table 1: **Percent accuracies for zero-shot image classification**. For CLIP where two numbers are reported, the accuracy correspond to two settings: downsizing the images to 64x64 and then resizing the images up to 224x224 (so CLIP does not have an advantage in input resolution over the 64x64 Imagen model) and resizing the images directly to 224x224 (so CLIP has the advantage of higher resolution). Variances in accuracy are <1% across different random seeds. The top set of results are on low-resolution datasets (which is why SD performs poorly).

while SD generally performs poorly (see Figure 6 in the appendix). This demonstrates that Imagen's areas of strength in generation carry over to downstream tasks and suggests that classification on OCR datasets could be used as a quantitative metric to study a model's text-generation abilities. SD generally performs poorly on the low-resolution datasets, perhaps because it is only trained on high-resolution images.[6] To better understand how much low-resolution is to blame, we evaluated SD on ImageNet after down-sampling the images to $32 \times 32$ and $64 \times 64$ resolution. SD's accuracy drops from $61.9\%$ to $15.5\%$ and $34.6\%$ respectively. The next five datasets use higher-resolution images. For some of these, taking advantage of CLIP's higher input resolution substantially improves results. SD performs comparably to Imagen on all these datasets (although of course it has an advantage in terms of input resolution).

Due to our reduced-size evaluation sets, variances in accuracy on zero-shot classification tasks across different random splits are roughly $\pm 0.4\%$ for CLIP, $\pm 0.7\%$ for Imagen, and $\pm 0.6\%$ for Stable Diffusion. The diffusion models have higher variance due to the inherent randomness in noising images (while CLIP is deterministic). Overall, we are not interested in small accuracy differences anyway, as the comparison between models is non-direct in various ways; instead we are trying go get a broad understanding of the models' abilities.

To our knowledge, these results are the first instance of a generative model achieving classification accuracy competitive with strong transformer-based discriminative methods. Lastly, we note that our method relies on the model being highly sensitive to text prompt, which we observe qualitatively in Figure 3.

| **Imagen** | **Stable Diffusion** | **CLIP** | **ViT-22B** | **ResNet50** (supervised) |
|---|---|---|---|---|
| **84.4** | 72.5 | 51.6 | 68.7 | 79 (top-5) |

Table 2: Percent shape accuracy for zero-shot classification on the Cue-Conflict Imagenet dataset.

## 4.2 Robustness to Shape-Texture Conflicting Cues

We next study diffusion models' robustness to presence of texture cues in images by evaluating their performance on the Cue-Conflict dataset from Geirhos et al. (2019). The dataset consists of Imagenet images altered to have a shape-texture conflict. While (for example) changing an image of a cat to have the texture of an elephant skin doesn't confuse humans, it could cause a model to classify the image as an elephant. Geirhos et al. (2019) showed that CNNs trained on Imagenet were strongly biased towards recognising textures rather than shapes, which is in stark contrast to human behavioural evidence.

---

[6]while low-resolution images were incorporated in CLIP's training, doing so with SD would run the risk of the model producing blurry images during generation

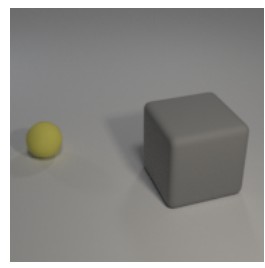

**Control tasks** test if the model can identify basic image features by scoring an attribute in the image against one not present. For example:
Shape: A sphere. vs. A cylinder. Color: A gray object. vs. A red object.,

**Binding tasks** test if the model binds a given attribute to the correct object. For example: Color|Shape: A yellow sphere. vs. A gray sphere.
Color|Position: On the right is a gray object vs. On the right is a yellow object.

**Pair binding tasks** are easier binding tasks where information about both objects is provided. For example:
Shape,Size: A small sphere and a large cube. vs. A large sphere and a small cube.
Color,Size: A small yellow object and a large gray object. vs. A large yellow object and a small gray object.

Figure 4: Examples of the synthetic-data attribute binding tasks. We explored more sophisticated prompts than in the figure (e.g., "A blender rendering of two objects, one of which is a yellow sphere."), but they didn't substantially change results.

We test Imagen's and Stable Diffusion's robustness to detecting shapes in presence of texture cues by using the same setting for classification as in Section 4.1. We report shape accuracy which is the percentage of images for which the model predicts the image's shape correctly in Table 2. We compare Imagen and SD with CLIP, the recently proposed ViT-22B model (Dehghani et al., 2023) which was trained on JFT (Sun et al., 2017) extended to 4B images (Zhai et al., 2022) and fine-tuned on Imagenet, and a (not zero-shot) supervised ResNet50 model trained on the training set. Imagen and SD comphrehensively outperform all previous methods on this dataset. Imagen further achieves an accuracy of 84.4% outperforming SD, CLIP, and ViT-22B by more than 12%, 30% and 15% respectively, and the top-5 accuracy performance of the supervised ResNet50 model by 5%.

We believe that the denoising process of the diffusion model is critical in removing the texture bias commonly observed in supervised models, making it robust to presence of textural cues. These findings are in line with Nie et al. (2022), who achieve state-of-the-art adversarial robustness through denoising adversarial examples with a diffusion model. We further qualitatively confirm this in Figure 3 in the appendix which depicts example images from this dataset and Imagen's result after denoising those images conditioned on text prompts with both the correct and incorrect shape class.

## 4.3 Evaluating Attribute Binding on Synthetic Data

We have shown that Imagen and SD perform comparably to CLIP at zero-shot classification, and much better than CLIP at disregarding misleading textural cues. Do diffusion models have additional capabilities that are difficult to obtain through contrastive pre-training? We hypothesize that one such area may be in compositional generalization, and specifically compare the models at attribute binding. Text-to-image generative models have shown emergent compositional generalization at large enough scale, being able to combine diverse concepts to handle prompts such as "a chair shaped like an avacado" (Ramesh et al., 2021). Attribute binding is a key piece of compositional reasoning, as it enables the understanding and integration of multiple concepts into a coherent whole. For example in the statement "a yellow sphere and a gray cube" we understand the sphere is yellow and the cube is gray, not the other way around. While previous work has examined attribute binding in text-to-image models by examining model generations (Nichol et al., 2021; Yu et al., 2022; Feng et al., 2023), our diffusion model classifier offers a way of more precisely studying the question quantitatively. We hope in the future, this type of study enabled by our method will be useful for comparing other abilities of generative image models at a fine-grained level.

**Dataset Construction:** We use synthetic images similar to Lewis et al. (2022), where images are generated based on the CLEVR (Johnson et al., 2017) visual question answering dataset. CLEVR images contain various object (cubes, cylinders, and spheres) with various attributes (different sizes, colors, and materials). A modified version of the CLEVR rendering script is used to generates images containing two objects of different shapes. From these images, we construct binary classification tasks of 1000 examples each; see Figure 4 for more details and examples. We follow the same setup as in the classification evaluation, using the $64 \times 64$ Imagen model and $512 \times 512$ Stable Diffusion model with heuristic timestep weighting and largest public CLIP model (with full-resolution inputs).

| Tasks | Imagen | Stable Diffusion | CLIP |
|---|---|---|---|
| Shape (control task) | **85** | **91** | **91** |
| Color (control task) | **96** | **85** | **94** |
| Shape,Color / Shape\|Color / Color\|Shape | **100** / **66** / **73** | **85** / **65** / **59** | 54 / 52 / 53 |
| Shape,Size / Shape\|Size / Size\|Shape | **99** / 48 / 51 | **63** / 48 / 52 | 52 / 51 / 50 |
| Shape,Position / Shape\|Position / Position\|Shape | **74** / 51 / 52 | 49 / 50 / 50 | 50 / 48 / 51 |
| Color,Size / Color\|Size / Size\|Color | **86** / 54 / 54 | **59** / 52 / 48 | 48 / 51 / 48 |
| Color,Position / Color\|Position / Position\|Color | **72** / 49 / 49 | 53 / 51 / 49 | 49 / 50 / 49 |
| Size,Position / Size\|Position / Position\|Size | **69** / 50 / 54 | 54 / 49 / 49 | 51 / 50 / 48 |

Table 3: Percent accuracy for models on zero-shot synthetic-data tasks investigating attribute binding. Bold results are significant ($p < 0.01$) according to a two-sided binomial test. CLIP is unable to bind attributes, while Imagen and SD sometimes can.

**Results:** Scores for Imagen, SD, and CLIP at these tasks is shown in Table 3. On the control tasks, all the models are able to identify shapes and colors that occur in the image with high accuracy. Imagen is slightly worse at shape identification; we find most of these are due to it mixing up "cylinder" and "cube" when the objects are small. Mistakes in color recognition generally occur when the distractor color is similar to the true color or to the color of the other object in the image (e.g. the distractor color is blue and there is a large cyan object in the image).

While CLIP can recognize image attributes, it performs no better than random chance for the attribute binding tasks. This result shows it is unable to connect attributes to objects and is consistent with the prior study from Subramanian et al. (2022). In contrast, Imagen can perform (at least to some extent) the pair binding tasks, and does better than chance on the Shape|Color and Color|Shape tasks. SD cannot perform the positional tasks, but can perform shape/color binding.

Part of Imagen's advantage might be in its text encoder, the pre-trained T5 (Raffel et al., 2020) model. Saharia et al. (2022b) find that instead using CLIP's text encoder for Imagen decreased its performance on generations involving specific colors or spatial positions. Similarly, Ramesh et al. (2022) find that DALLE-2, which uses a CLIP text encoder, is worse at attribute binding than GLIDE, which uses representations from a jointly-trained transformer processing the text. An advantage of the diffusion models over CLIP is their use of cross attention to allow interaction between textual and visual features. A visual model without completely independent text and image encoders such as LXMERT (Tan & Bansal, 2019) or CMA-Clip (Liu et al., 2021) might perform better, but of course these models come with the added compute cost of having to jointly process all image-text pairs with the model instead of embedding the text and images separately.

One mistake we observed frequently in Color|Shape with Imagen is it preferring the color of the larger object in the image; e.g. scoring "A gray sphere" over "A yellow sphere" in Figure 4. We hypothesize that it is helpful for denoising at high noise levels when the text conditioning provides the color for a large region of the image, even when the color is associated with the wrong shape. In the pair task, the full color information for both objects is always provided, which avoids this issue, and perhaps explains why accuracies at pair tasks are much higher.

# 5   Conclusion and Future Work

We have proposed a method that enables diffusion models to be used as zero-shot classifiers and developed ways of improving its efficiency to make it usable. Our experiments demonstrate strong results on image classification. Furthermore, we show Imagen and Stable Diffusion are remarkably robust to misleading textures, achieving state-of-the-art results on cue-conflict dataset. While existing analysis of diffusion models usually studies generated images qualitatively, our framework provides a way of quantitatively evaluating text-to-image generative models through evaluating them on controlled classification tasks. We showcase this through our study on attribute binding, where we find that diffusion models are sometimes able to bind attributes while CLIP does not appear to have this ability. Similar experiments could be used in the future to study other properties of pre-trained diffusion models, such as toxicity or bias.

Our paper is complementary to concurrent work from Li et al. (2023), who use Stable Diffusion as a zero-shot classifier and explore some different tasks like relational reasoning. While their approach is similar to ours, they perform different analysis, and their results are slightly worse than ours due to them using a simple hand-tuned class pruning method and no timestep weighting.

We hope our findings will inspire future work in using text-to-image diffusion models as foundation models for tasks other than generation. One direction is fine-tuning diffusion models on downstream tasks; given the strong zero-shot performance of Imagen and Stable Diffusion, a natural next step is evaluating them after further supervised training. As models become larger, another key question for further study is how do the scaling laws (Hestness et al., 2017; Kaplan et al., 2020) of contrastive vs generative pre-training compare. Additionally, we are interested in applying our analysis to other generative models to study to what extent our results are a consequence of generative pre-training generally compared to diffusion pre-training specifically.

Ultimately, our method does not produce a practical classifier, as it requires substantial compute when scoring many classes. Instead, we see the main value of this work is in revealing more about the abilities of large pre-trained diffusion models and providing a method for enabling future fine-grained studies of diffusion model abilities. In total, our results suggest that generative pre-training may be a useful alternative to contrastive pre-training for text-image self-supervised learning.

## Acknowledgements

We thank Kevin Swersky, Mohammad Norouzi, and David Fleet for helpful discussions and feedback, Martha Lewis and Ellie Pavlick for answering our questions about their CLIP attribute binding experiments, and Robert Geirhos for answering our questions about Stylized Imagenet and ViT-22B. We also thank the anonymous reviewers for their comments and suggestions.

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

# A  Model Details.

**Imagen details:** Imagen is a text-conditioned diffusion model that comprises of a frozen T5 (Raffel et al., 2020) language encoder that encodes an input prompt into a sequence of embeddings, a $64 \times 64$ image diffusion model, and two cascaded super-resolution diffusion models that generate $256 \times 256$ and $1024 \times 1024$ images. Unlike Stable Diffusion, it operates directly on pixels instead of in a latent space. For our experiments, we use the $64 \times 64$ model which has 2B parameters and is trained using a batch size of 2048 for 2.5M training steps on a combination of internal datasets, with around 460M image-text pairs, and the publicly available Laion dataset (Schuhmann et al., 2021), with  400M image-text pairs.

**Stable Diffusion details**. We use Stable diffusion v1.4 which is a latent text-to-image diffusion model. It uses the pre-trained text encoder from CLIP to encode text and a pre-trained variational autoencoder to map images to a latent space. The model has 890M parameters and takes 512x512-resolution images as input. It was trained on various subsets of Laion-5B, including a portion filtered to only contain aesthetic images, for 1.2M steps using batch size of 2048.

**CLIP details:**  CLIP encodes image features using a ViT-like transformer (Dosovitskiy et al., 2021) and uses a causal language model to get the text features. After encoding the image and text features to a latent space with identical dimensions, it evaluates a similarity score between these features. CLIP is pre-trained using contrastive learning. Here, we compare to the largest CLIP model (with a ViT-L/14@224px as the image encoder). The model is smaller than Imagen (400M parameters), but is trained for longer (12.8B images processed vs 5.B). While Imagen was trained primarily as a generative model, CLIP was primarily engineered to be transferred effectively to downstream tasks.

# B  Weighting Functions Details.

**Learned Weighting Function:**    While for most experiments we use a heuristic weighting function for $\boldsymbol{w}_t$, we also explored learning an effective weighting function (although this is not truly zero-shot). To do this, we aggregate scores for each image $\boldsymbol{x}$ and class $\mathsf{y}_k$ into 20 buckets, with each bucket covering a small slice of timestep values:

$$\boldsymbol{b}_i(\boldsymbol{x}, \mathsf{y}_k) = \mathbb{E}_{\epsilon, t \sim \mathcal{U}[0.05i, 0.05(i+1)]} \|\boldsymbol{x} - \tilde{\boldsymbol{x}}_{\boldsymbol{\theta}}(\boldsymbol{x}_t, \boldsymbol{\phi}(\mathsf{y}_k), t)\|_2^2$$

where we estimate the expectation with Monte Carlo sampling (typically around 100 samples). We then learn a 20-feature linear model with parameters $[\boldsymbol{v}_0, ..., \boldsymbol{v}_{19}]$ over these buckets:

$$p_{\boldsymbol{v}}(y = \mathsf{y}_k | \boldsymbol{x}) = \frac{\exp(\sum_{i=0}^{19} -\boldsymbol{v}_i \boldsymbol{b}_i(\boldsymbol{x}, \mathsf{y}_k))}{\sum_{\mathsf{y}_j \in [\mathsf{y}_K]} \exp(\sum_{i=0}^{19} -\boldsymbol{v}_i \boldsymbol{b}_i(\boldsymbol{x}, \mathsf{y}_j))}$$

trained with standard maximum likelihood over the data. At test-time we use the weighting

$$\boldsymbol{w}_t = \boldsymbol{v}_{\lfloor t/0.05 \rfloor}$$

We generally found that (1) learned weighting functions are pretty similar across datasets, and (2) the weighting functions are transferable: the $\boldsymbol{v}$s learned on one dataset get good accuracy when evaluated on other ones. On average, learned weights produced around 1% higher accuracy on zero-shot classification tasks, but we omitted the results from the main paper because using learned weights is not truly zero-shot.

**Comparison of Weighting Functions.**    We compare the learned weighting functions with several heuristic functions on the Caltech101 dataset. We chose Caltech101 because it is high-resolution (SD performs poorly on low-resolution datasets), contains a diversity of image classes, was not used when we developed the heuristic weighting function, and only has 100 classes, so it is much faster to evaluate models on than ImageNet. We compare the following functions:

- **VDM**: $\boldsymbol{w}_t = \text{SNR}'(t)$, the derivative of the signal to noise ratio with respect to $t$. This weighting scheme from Variational Diffusion Models (Kingma et al., 2021) directly trains the model on a lower bound of the likelihood.
- **Simple**: $\boldsymbol{w}_t = \text{SNR}(t)$. This "simple" loss from Ho et al. (2020) results in a model that produces better images according to human judgements and FID scores, even though it results in worse likelihoods.

| Weighting | Imagen | Stable Diffusion |
|-----------|--------|------------------|
| VDM | 62.0 | 71.9 |
| Simple | 56.1 | 71.4 |
| Heuristic | 68.9 | 73.0 |
| Learned | 70.2 | 73.1 |

Table 4: Percent accuracy for models on Caltech101 with different weighting schemes

- **Heuristic**: $w_t = \exp(-6t)$. Our hand-engineered weighting function; we found this by searching for a simple weighting function that works well on CIFAR-100, although we found empirically it generalizes very well to other datasets.

- **Learned**: Learning an effective weighting function on a held-out set of examples as described above.

Results are shown in Table 4. The heuristic weighting function outperforms Simple and VDM for both models. Interestingly, SD appears to be more robust to the choice of weighting function than Imagen. Mechanistically, the reason is that "Simple" and "VDM" weighting put more weight on earlier timesteps than "Heuristic" and Imagen tends to be an inaccurate classifier at very small noise levels. We intuitively believe this is a consequence of pixel vs latent diffusion. The learned weighting only does slightly better than heuristic weighting despite not being truly zero-shot. We found similar results to hold on other datasets.

## C   Details on Attribute Binding Tasks and Prompts

We use the relational dataset from Lewis et al. (2022) for the attribute binding experiments. Each image consists of two objects of different shapes and colors; for tasks involving size we filter out examples where both objects are the same size. Each image contains two objects with different attributes shape $\in$ {cube, sphere, cylinder}, color $\in$ {blue, cyan, blue, brown, gray, green, purple, red, yellow}, size $\in$ {small, large}, and position $\in$ {left, right}.

Given a task (e.g. Shape|Size), we construct a task-specific description for an object as follows:

> *"On the {*position*} is a "* if Position tasks else *"A "*+
>
> "size " if Size task else ""+
>
> "color " if Color task else ""+
>
> "shape." if Shape task else *"object."*

For recognition and binding tasks, we randomly select one of the two objects in the image to be the positive example and then use its description as the positive prompt. For pair tasks, we join the descriptions for both objects together with "and" (removing the period from the first description and lowercasing the second one) for the positive prompt.

To construct a negative example for recognition tasks, we replace the positive attribute with a random attribute not in the image. For binding tasks, we replace one of positive description's attributes with the other object's attribute (e.g., for Shape|Color, we replace shape).

For pair tasks, there is a choice in how the two objects are ordered (e.g. "On the left is a cube and on the right is a sphere" vs "On the right is a sphere and on the left is a cube". We follow the preference of stating the leftmost position/shape/color/size first in that order. For example, this means we will always start with "On the left..." rather than "On the right...". Similarly, the negative example for Color,Size in Figure 4 is "A large yellow object and small gray object" rather than "A small gray object and a large yellow object" because we prefer to first put the leftmost color over the leftmost size.

We experimented with a variety of other prompts, but found none to work substantially better than these simple ones.

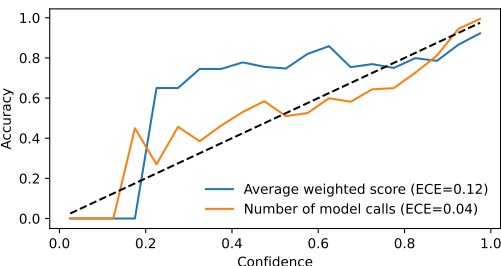

Figure 5: Model reliability diagram comparing confidence measures of Imagen on CIFAR-100. The number of model calls used in Algorithm 1 produces better-calibrated confidences than using the actual scores for different classes.

## D   Calibration

It is desirable for classifiers, especially when used in the zero-shot setting with possibly out-of-domain examples, to be well calibrated. In other words, if a classifier predicts a label $\tilde{y}_i$ with probability $p$, the true label should be $\tilde{y}_i$ roughly $100 \cdot p\%$ of the time. However, the diffusion model classifier does not directly produce probabilities for classes. While $p(y_i = y_k | \boldsymbol{x}_i)$ should roughly be proportional to the expectation in Equation (2) when exponentiated (i.e. we can apply a softmax to the average weighted scores to get probabilities), in practice our estimates of the expectations are very noisy and do not provide well-calibrated scores.

We propose a simple alternative that takes advantage of early pruning: we use the total number of diffusion model calls used for the image as a calibration measure. The intuition is that a harder example will require more scores to determine the $\arg\min$ class with good statistical significance.

More details on the two calibration methods are below:

**Temperature-scaled raw scores.**    We use $s_{y_k}(\boldsymbol{x})$ to denote the weighted average squared error for class $y_k$ on image $\boldsymbol{x}$, i.e., the Monte-Carlo estimate for the re-weighted VLB in equation 2. We turn these scores into an estimated probability by applying a softmax with temperature:

$$p_\theta(y = y_k | \boldsymbol{x}) = \frac{\exp(-s_{y_k}(\boldsymbol{x})/\tau)}{\sum_{y_j \in [y_K]} \exp\left(-s_{y_j}(\boldsymbol{x})/\tau\right)}$$

Note that this approach requires good score estimates for each class, so it is not compatible with the class pruning method presented in Section 3.1.

**Platt-scaled number of scores.**    Our other confidence method relies on the total number of scores needed to eliminate all other classes as candidates. Let $\tilde{y}(\boldsymbol{x})$ denote the predicted class for example $\boldsymbol{x}$ and $n(\boldsymbol{x})$ be the total number of calls to $\tilde{\boldsymbol{x}}_\theta$ used to obtain the prediction when running Algorithm 1. Then we estimate

$$p_\theta(y = \tilde{y}(\boldsymbol{x}) | \boldsymbol{x}) = \text{sigmoid}(-n(\boldsymbol{x})/\tau + \beta)$$

We learn $\tau$ (and $\beta$ for Platt scaling) on a small held-out set of examples.

We show reliability diagrams (DeGroot & Fienberg, 1983) and report Expected Calibration Error (Guo et al., 2017) (ECE) for the methods in Figure 5. Using a small held-out set of examples, we apply temperature scaling (Guo et al., 2017) for the score-based confidences and Platt scaling (Platt et al., 1999) for the number-of-scores confidences, (see Appendix D for details). Number of scores is fairly well-calibrated, showing it is possible to obtain reasonable confidences from diffusion model classifiers despite them not providing a probability distribution over classes.

## E   Imagen's Super-resolution Models

Imagen is a cascaded diffusion model (Ho et al., 2022) consisting of a $64 \times 64$ low-resolution model and two super-resolution models, one that upsample the image from $64 \times 64$ to $256 \times 256$ and one

Prompt: A photo of the number 8.

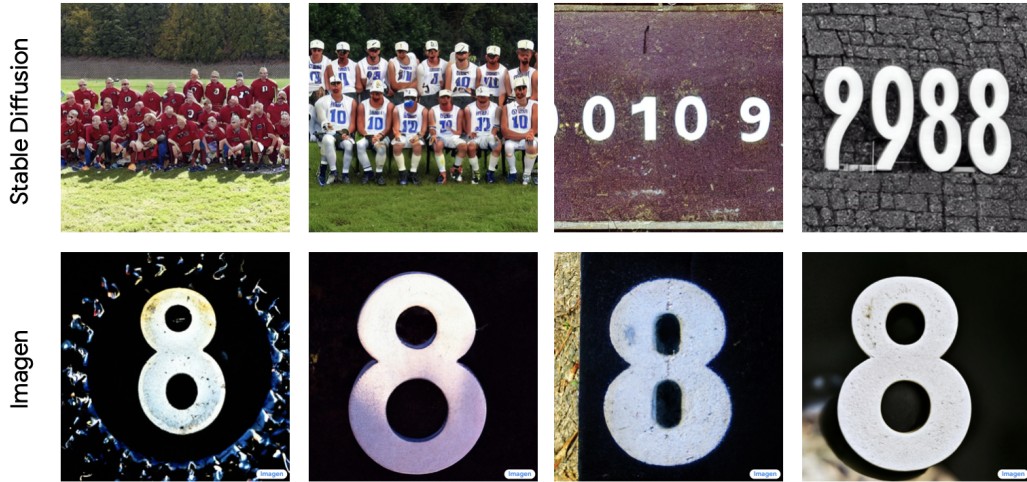

Figure 6: Example generated images from Stable Diffusion and Imagen for generating photos with text.

that upsamples from $256 \times 256$ to $1024 \times 1024$. However, we found only the $64 \times 64$ model to work well as a zero-shot classifier. The super-resolution models condition on a low-resolution input image, which means they denoise effectively regardless of the input prompt and thus aren't as sensitive to the class label. For example, unlike with Figure 3, high-resolution denoising with different text prompts produces images imperceptibly different to the human eye because they all agree with the same low resolution image. Imagen's super-resolution models are trained with varying amount of Gaussian noise added to the low-resolution input image (separate from the noise added to the high-resolution image being denoised). We were able to alleviate the above issue somewhat by using a large amount of such noise, but ultimately did not achieve very strong results with the high-resolution models. For example, the $64 \times 64$ to $256 \times 256$ model achieves an accuracy of 16.1% on ImageNet.

We further experimented with combining the low-resolution model's scores with the $64 \times 64$ to $256 \times 256$ model's. To do this, we used the learned weighting scheme detailed in Appendix B, but with learning 40 weights: 20 for the low resolution model and 20 for the super-resolution model. However, we found the learned weighting scheme put almost no weight on the super-resolution model's scores, and did not perform significantly better than the low-resolution model did on its own.

