# OpenReview forum: "Text-to-Image Diffusion Models are Zero Shot Classifiers"
_NeurIPS.cc/2023/Conference — NeurIPS 2023 spotlight_

### Official Review · Reviewer_edqQ · 2023-07-04

**Soundness:** 3 good
**Presentation:** 3 good
**Contribution:** 3 good
**Rating:** 7
**Confidence:** 4

**Summary:**

This paper uses the text-to-image diffusion models as zero-shot classifiers. It proposes to compute a subset of the full scores matrix to be more efficient. It proves Imagen and Stable Diffusion have good zero-shot performance and are robust to misleading textural cues.

**Strengths:**

1. It is novel to use pre-trained diffusion models as zero-shot classifiers.
2. Building a scores matrix with each label prompt is reasonable for me.
3. The efficiency is very important in this setting. The author well discussed this point in section 3.1.
4. It is interesting to see the generalization ability of large pretrained model leads to robustness to shape-texture conflicting cues.
5. Empirical results are sufficient to support the conclusion.

**Weaknesses:**

1. It lacks some theoretical discussion about why diffusion models can perform well in classification.
2. Classification is a simple task. I am not sure whether it is too easy for a large model.

**Questions:**

1. Is it possible to extent this work to various downstream tasks such as detection or segmentation?
2. In large datasets like ImageNet, there are many kinds of sub-classes such as many types of foxes. In this case, is it reasonable to use the diffusion model since they are very similar in noisy images?

**Limitations:**

Please refer to "Questions".

---

> ### Author Rebuttal · Authors · 2023-08-10
>
> Thank you for the helpful comments and suggestions. We address your questions and concerns below:
>
> - **Theoretical discussion**: Generally, the idea of using a generative model as a classifier is a fairly old and well-studied idea (e.g. “On Discriminative vs. Generative Classifiers: A comparison of logistic regression and naive Bayes” NeurIPS 2001). Our main contribution is in showing how to efficiently use a diffusion model this way and presenting empirical results comparing them with contrastive zero-shot models.
>
> - **Classification is a simple task**: We think image classification tasks certainly can be very challenging; for example CLIP gets around 50% on cue-conflict imagenet and does no better than chance in our attribute binding experiments.
>
> - **Q1 Extension to other downstream tasks**: There have been several recent works that have used pre-trained text-image diffusion models (either as zero-shot or through their representations) for segmentation and detection e.g. https://arxiv.org/abs/2303.04803, https://arxiv.org/abs/2211.13224, https://arxiv.org/abs/2303.02153 etc. However, the method proposed in the paper would not work directly for many downstream tasks because it scores images based on the text prompt rather than predicting some structured output.
>
> - **Q2 Fine-grained image classification**: We agree with the intuition that distinguishing sub-classes would be hard when there is lots of noise. However, our method uses many noise levels and we expect the scores at low noise-levels will be more discriminative for distinguishing similar classes. Indeed, our heuristic weighting function puts more weight on lower noise levels.

---

### Official Review · Reviewer_13fF · 2023-07-06

**Soundness:** 3 good
**Presentation:** 3 good
**Contribution:** 3 good
**Rating:** 6
**Confidence:** 4

**Summary:**

The paper presents a new method that utilizes generative models as image classifiers and initiates explorations of notable, open-source models using this approach. Beginning with recognized image datasets, the researchers examine the models and evaluate the scores they have achieved. Several experiments are conducted to profile the behavior of these models, such as determining their optimal operating resolutions and dataset types. The paper assesses these models' capabilities by evaluating dataset competencies, such as pointing out the prominence of the Imagen model with the MNIST dataset due to its strong text generation skills.

Further characterizations are made on the models' ability to handle shape-texture conflicting cues using the Cue-Conflict dataset. The paper suggests that generative models, utilized through their proposed method for recognition tasks, exhibit more robustness compared to traditional ConvNets.

Lastly, the paper explores how these models perform in attribute binding tasks. The researchers report that while the CLIP model's performance is near random, both the Stable Diffusion (SD) and Imagen models show promising potential in these tasks.

**Strengths:**

The strengths of this paper shine through in several ways. Firstly, the paper explains the research and results in a way that's easy to understand. This clear writing helps to make complex ideas accessible to a wider audience.

The paper's thorough exploration of generative models also sets this work apart. They dive deep into understanding how these models behave. This includes looking at how well the models can handle images where shapes and textures don't match, as well as how they can connect attributes together. These insights are not just interesting, but they also add to our understanding of how these models work.

Another key strength of this paper is the smart improvements the paper makes. For instance, they use a technique called timestep weighting to improve how the models classify images. This technique is a smart way to reduce the impact of noise, making the models more reliable as they handle larger timesteps if they are sampled. Such improvements amplify the significance of the paper.

**Weaknesses:**

# Pre-rebuttal
While the paper continually emphasizes the intention to introduce a method for using generative models as classifiers, there's a significant overlap with the methodology presented in "Your Diffusion Model is Secretly a Zero-Shot Classifier", which also uses the Stable Diffusion generative model as a classifier. Despite the similar scoring mechanisms, they attribute their improved results to the application of timestep weighting and a more efficient class pruning method. This does not, however, fully offset the lack of originality due to the similarity of their method with the cited work. Furthermore, by choosing not to pursue enhancements such as prompt engineering, the authors have seemingly missed an opportunity to establish a stronger benchmark for future studies. This decision, coupled with the questions regarding the paper's originality, could limit the overall significance and potential impact of their work.

Certain parameters used in the paper, such as the cutoff_pval, are not adequately examined for their effects, making them appear as arbitrary choices or 'magic numbers'. Similarly, the weighting function used remains largely unexplored and unexplained, leaving a gap in understanding its impact on the results.

The paper provides comparisons for their efficiency improvements, including shared noise and class pruning methods. However, the paper falls short in determining the peak accuracy reach for the vanilla method, leaving an element of uncertainty and a lack of thorough comparison between the methods employed.



**Questions:**

1. L271: few-shot classification?
2. Why not use more commonly used zero-shot (or few-shot) learning datasets such as CUB, AwA or miniImageNet?
3. Why not compare against strong (even if not sota) ZSL baselines with various benchmarks?
4. How do you tune the hyper-parameters? Setting any HP wrt. test performance is a violation of ZSL protocol. Can you describe a reproducible HP tuning procedure?
5. Regarding the compositional generalization experiments: why not use existing compositional zsl benchmarks and compare against existing methods?


**Limitations:**

Most limitations are listed above. No additional (especially societal) limitation to be reported here.

---

> ### Author Rebuttal · Authors · 2023-08-09
>
> Thank you for the helpful comments and suggestions. We address your questions and concerns below:
>
> - **Overlap with another paper**: As we state in the conclusion, “Your Diffusion Model is Secretly a Zero-Shot Classifier” is concurrent work. It was released on arxiv (but not in a peer-reviewed venue) around a month before NeurIPS submission deadline. In fact, our method and results were submitted and presented in workshops from the end of January 2023 and were thus conducted before the other work (to preserve the sanctity of double blind reviewing, we can not provide exact details on this). It is our understanding that the existence of a recent non-peer-reviewed pre-print that is similar to the work under submission should not affect novelty/originality judgements.
>
> - **Not pursuing enhancements such as prompt engineering**: We actually extensively explored different prompts, but found none to significantly outperform simple prompts like “a photo of a __.” (e.g. see lines 207-209).
>
> - **Weighting function unexplored**: We did explore the weighting function in detail, and included those results and discussion in the supplementary section due to space constraints (see Appendix B and its reference in the main text in Line 127).
>
> - **Hyperparameter selection**: The choice of *min_scores, max_scores*, and *cutoff_pval* are simply reasonable choices that keep the method efficient to run, not crucial “magic numbers.” Different choices of these parameter values will make the method use a different amount of compute (with less compute leading to noisier estimates of the best-scoring classes). However, different choices of these values do not change the actual behavior of the model in expectation because the method is always estimating the same argmin (eq 2). Furthermore, we selected these values and weighting function using CIFAR-100 (see section 3, Lines 116-127) but not on any of the many other datasets we experimented on. We have been careful to preserve true ZS classification protocol in our experiments.
>
> - **Peak accuracy for vanilla method**: The “efficient” and “vanilla” methods are different ways of estimating exactly the same argmin (eq 2 in the paper), so they have the same peak accuracy given enough compute. However, the methods do have different efficiencies, which is shown in Figure 2.
>
> - **Q1 L271**: Good catch! That should be “zero-shot” classification.
>
> - **Q2 Other datasets**: CUB, AwA, and miniImageNet are mostly used for few-shot classification, which we don’t explore in our paper. Instead, we use datasets from the CLIP paper, which have become pretty standard for evaluating zero-shot classification (e.g. used to evaluate ALIGN, BASIC, and LiT)
>
> - **Q3 Strong baselines**: We think CLIP is a pretty strong baseline for zero-shot classification. While there exist stronger models, many are either not open source (e.g. CoCa) or are pre-trained on classification-like data (e.g. LiT). Additionally, most subsequent zeros-shot methods are based on CLIP. We are not sure which other baseline the reviewer had in mind.
>
> - **Q4 Hyperparameters**: See our “hyperparameter selection” bullet point above. Briefly, we selected the hyperparameter values and weighting function using CIFAR-100 (see section 3, Lines 116-127), but not on any of the many other datasets we experiment on.
>
> - **Q5 Why not use an existing benchmark**: We are using an existing benchmark from Lewis et al. (https://arxiv.org/abs/2212.10537).

---

> > ### Comment · Reviewer_13fF · 2023-08-16
> >
> > I’d like to thank the authors for their detailed response.
> >
> > * The authors are right that “Your Diffusion Model is Secretly a Zero-Shot Classifier” is indeed arxiv-only and apparently quite recent (1.5 months before NeurIPS deadline), so it can be ignored following the NeurIPS’23 policy. It is an unintentional mistake on my side. However, as a suggestion, I believe it will be beneficial for everyone if the paper acknowledges the arxiv paper as concurrent recent work and discuss the (dis)similarities briefly.
> >
> > * Prompts: thanks, please add a brief summary on your explorations on prompt enhancements in the paper.
> >
> > * HPs: thanks for the clarification. I find the explanation given the paper a bit cryptic. The answer in the rebuttal is much more clear. I’d strongly suggest to revise the explanation in the paper to improve clarity.
> >
> > * Datasets: traditional ZSL papers (without web-scale training and more “clean” protocols) actually predominantly do use CUB & AwA (but not miniImageNet indeed), I disagree with the response, but existing evaluations are acceptably strong.
> >
> > * Other points: thanks for the clarifications & pointers, they’re sufficient & clear.
> >
> > While some (minor) revisions based on rebuttal will be pending for the camera-ready version, following the important clarifications I have increased my rating to ‘weak accept’.

---

> > > ### Author Response · Authors · 2023-08-17
> > >
> > > Thank you for the comments and feedback. We will make the requested clarifications in the revised paper.

---

### Official Review · Reviewer_yeML · 2023-07-06

**Soundness:** 3 good
**Presentation:** 3 good
**Contribution:** 3 good
**Rating:** 6
**Confidence:** 4

**Summary:**

This study explores the potential of text-to-image diffusion models as zero-shot classifiers. The models show competitive performance with CLIP on zero-shot image classification datasets and excel in shape/texture bias tests and attribute binding. The findings suggest that generative pre-training should be considered as a compelling alternative for vision-language tasks.

**Strengths:**

1. The authors thoroughly investigate the zero-shot classification capabilities of text-to-image diffusion models through extensive experiments, covering both standard and challenging benchmarks.

2. Additionally, the authors compare the classification performance of Stable Diffusion (SD) and Imagen, revealing intriguing results that suggest diffusion models trained on original images exhibit superior generalization compared to those trained on the latent space of VAE.

3. The authors offer a range of practical and effective techniques to accelerate the classification process, providing valuable insights for improved efficiency.

**Weaknesses:**

1. While the authors present various strategies to speed up the classification process, it remains significantly slower compared to traditional classification models.

2. It would be beneficial to include a comparison with other multimodal models like BLIP.

3. There appears to be a disconnect between lines 192 and 193, as the introduction of Imagen is only mentioned in that paragraph.

4. Considering that CLIP is trained with 400M text-image pairs and SD is trained on Laion-5B, a direct comparison may not be entirely fair. It would be interesting to see the performance gap between SD and CLIP when both models are trained on the same amount of data.

**Questions:**

See weaknesses.

---

> ### Author Rebuttal · Authors · 2023-08-09
>
> Thank you for the helpful comments and suggestions. We address your questions and concerns below:
>
> - **Runtime**: As we say in the paper, the method does not produce a very practical classifier. However, we believe it still has a lot of value for illuminating what kinds of visual knowledge diffusion models learn, comparing diffusion model abilities on fine-grained tasks (such as attribute binding), and more generally comparing generative vs. contrastive pre-training.
>
> - **Comparison with other multimodal models**: We focused on CLIP as it is the most widely used zero-shot classifier and most subsequent methods (including BLIP) use a similar contrastive training method. As far as we know, BLIP is not often used as a zero-shot image classifier (although of course it is used for other tasks like captioning).
>
> - **Lines 192,193**: Sorry for the confusion: that sentence was meant to indicate which version of Stable Diffusion we are using, not suggest we are only using Stable Diffusion. We will rephrase it to avoid confusion.
>
> - **Different training datasets**: We completely agree, and hope in the future the community will release more comparable models. Please see the general response above for more discussion and our explanation on model comparisons. We also note that the dataset comparison between Stable Diffusion and CLIP isn’t entirely clear because (1) CLIP was trained for 32 epochs, while SD was trained for less than 1, and (2) SD was largely trained on an aesthetic subset of LAION-2B, which is substantially smaller than the full LAION-5B dataset.

---

### Official Review · Reviewer_LkNq · 2023-07-07

**Soundness:** 3 good
**Presentation:** 4 excellent
**Contribution:** 3 good
**Rating:** 6
**Confidence:** 4

**Summary:**

The paper inverts pre-trained text-to-image diffusion models by using bayes rule, and evaluates them over a variety of benchmarks. For the evaluation they use two diffusion models: Stable Diffusion and Imagen. They compare these models against CLIP-L/14. They show a variety of benchmarks where the diffusion model does better at classification than a standard SOTA discriminative model such as CLIP-L/14.

**Strengths:**

i) The paper does a very dense evaluation of their proposed method

ii) They give good analysis on why/where generative classifiers would be useful over their discriminative counterpart.

iii) The paper proposes weighted timesteps and sampling methods that improve the accuracy and speed of the classifier.

**Weaknesses:**

i) the analysis become a bit weak , as none of the models considered are trained on the same dataset.

ii) Ablations are in the supplementary and results are ablated only on one dataset, would be good to have atleast 2-3 diverse datasets (small/high resolution)

I have few qs that i have listed below.

minor -
Line 271 shouldn't it be zero-shot instead of few-shot?


**Questions:**

i) In Table 4 weighting seems to help significantly help Imagen however not so much for SD? why is that? what if u learned weighting for SD seperately and tried to generalize it to different datasets?

ii) Are any of the models trained on the same/similarish datasets (ViT22B vs Imagen)?
It would be good to clarify this in the paper.

iii) The paper raises resolution mismatch of the generative model (specifically Imagen), a big concern. Do the authors expect future generative models that are explicitly trained for classification to resolve this issue? If so how? as SD does train on higher resolution but doesn't get better performance than CLIP





**Limitations:**

Yes

---

> ### Author Rebuttal · Authors · 2023-08-09
>
> Thank you for the helpful comments and suggestions. We address your questions and concerns below:
>
>
> - **Different training datasets**: As we mentioned in the general response above, in this work, we focused on studying the capabilities of existing powerful models because training such models from scratch would require huge compute resources. However, many of the differences between models are striking enough that we are confident they point to fundamental differences in what is learned from their training objectives rather than differences in their training data. For example, diffusion models greatly outperform CLIP on robustness to shape-texture conflicts and attribute binding, and pixel-based diffusion substantially outperforms latent-space based diffusion models on OCR data like MNIST and SVHN (see Lines 231-236).
>
>
> - **Ablations**: As we state in line 536 of the supplementary materials, we did find similar results to hold across many datasets. We used our results on Caltech 101 as a representative example to support our findings, but we are happy to include numbers on other datasets in the revised version of the paper.
>
>
> - **Line 271**: Good catch – yes, that should say zero-shot.
>
>
> - **Q1 Learned weighting**: We did try learning the weighting for SD separately on Caltech101 (last row of Table 4 in Appendix B). We did not try applying the learned weights on other datasets because it barely improved over the heuristic weighting while being more complicated. We also found it interesting that SD is more robust to the choice of weighting scheme. Mechanistically, the reason is essentially that (1) “Simple” and “VDM” weighting put more weight on earlier timesteps than “Heuristic” and (2) Imagen tends to be an inaccurate classifier at very small noise levels. We intuitively believe this is a consequence of pixel vs latent diffusion.
>
>
> - **Q2 Models trained on similar datasets**: Imagen and Stable Diffusion are perhaps most similar in that both are trained on LAION (although Imagen uses some additional data). ViT22B is trained on JFT, which is less similar because it is a (semi-automatically labeled) classification dataset. We discuss training data for ViT on line 259 and for Imagen, SD, and CLIP in section A of the supplementary materials.
>
>
> - **Q3 Resolution mismatch**: We expect non-cascaded diffusion models such as latent diffusion models or simple diffusion (https://arxiv.org/abs/2301.11093) to avoid the resolution mismatches. We also think explicitly fine-tuning generative models for classification would improve results and would be an interesting future direction of research. While CLIP outperforms SD on most classification tasks, SD is better at attribute binding and the cue-conflict dataset, suggesting that the different pre-training methods may have different areas of strength rather than one being strictly better than the other. Intuitively, it makes sense that CLIP is better at standard image classification, as it was designed with transfer to classification tasks in mind while SD and Imagen were designed as image generators.

---

> > ### Comment · Reviewer_LkNq · 2023-08-14
> >
> > Thanks for the response.
> >
> > I'm wondering if the authors could comment or compare against LiT (https://openaccess.thecvf.com/content/CVPR2022/html/Zhai_LiT_Zero-Shot_Transfer_With_Locked-Image_Text_Tuning_CVPR_2022_paper.html)
> >
> > LiT uses a pre-trained language model to fine-tune discriminative image models. As one could potentially argue that the improvement in performance is primarily due to the use of Language model in Imagen that other discriminative models don't use.

---

> > > ### Author Response · Authors · 2023-08-14
> > > **Use of pre-trained LM and comparison with LiT**
> > >
> > > Stable Diffusion uses a frozen CLIP text encoder rather than a language model and exhibits similar behavior to Imagen (decent at zero-shot classification, excellent on cue-conflicted imagenet, better than chance on attribute binding), even though results are generally a bit worse. We therefore think CLIP vs. Stable Diffusion already is a good comparison for evaluating differences in image pre-training methods while keeping the text encoder identical. Along similar lines, [this paper](https://arxiv.org/pdf/2303.09769.pdf) shows that unconditional diffusion pre-training (without any text component) performs well when transferred to downstream tasks, offering additional evidence that the diffusion training is useful for representation learning aside from the choice of text model.
> > >
> > > We do agree that the improvement of Imagen over Stable Diffusion could be due to the more powerful text encoder, but we think LiT may not be a good comparison for looking into this more. The issue is that the LiT image encoder is pre-trained on JFT-3B, a semi-automatically labeled fine-grained image classification dataset, and then frozen. Since LiT's image encoder essentially sees classification data during pre-training, we don’t think using it is truly zero-shot, so the comparison would not be direct.

---

### Author Rebuttal · Authors · 2023-08-09

### **General response to all reviewers**

We thank all the reviewers for their helpful comments and suggestions.

Generally, reviewers (*LkNq* and *yeML*) had questions around the fairness of comparison between models trained on different datasets (such as CLIP and Stable Diffusion). In this work, we focused on studying the capabilities of existing powerful models because training such models from scratch would require huge compute resources. However, many of the differences between models are striking enough that we are confident they point to fundamental differences in what is learned from their training objectives rather than differences in their training data. For example, diffusion models greatly outperform CLIP on robustness to shape-texture conflicts and attribute binding, and pixel-based diffusion substantially outperforms latent-space based diffusion models on OCR data like MNIST and SVHN (see Lines 231-236).

Secondly, reviewers (*LkNq* and *13fF*) had questions regarding ablation studies for the weighting function. We studied the effects of different weighting functions in detail and have included this discussion in Appendix B.

Reviewers (*yeML* and *13fF*) also had questions regarding comparisons to other baselines. In our work, we focused on CLIP as it is the most widely used zero-shot classifier and most subsequent methods incorporate similar contrastive training methods. While stronger models than CLIP do exist, many are either not open source (e.g. CoCa) or are pre-trained on classification-like data (e.g. LiT). For all our experiments, we have compared to existing benchmarks like datasets used in the CLIP paper for image classification, benchmark proposed by Geirhos et.al 2021 for robustness to shape-texture bias and the benchmark proposed by Lewis et.al 2023 for attribute binding.

We address the further concerns and questions of each reviewer in detail below.

---

### Decision · Program_Chairs · 2023-09-21

**Decision:**

Accept (spotlight)

**Comment:**

The paper compares generative diffusion models (SD and Imagen) and discriminative image-text models such as CLIP on the task of zero-shot classification. After the rebuttal all reviewers recommend acceptance. As large-scale diffusion models are relatively new, their properties are not yet fully understood. This paper provides thorough analysis and insights and will be very valuable, especially for many downstream applications of these models.